# Iterative In Silico Screening for Optimizing Stable Conformation of Anti-SARS-CoV-2 Nanobodies

**DOI:** 10.3390/ph17040424

**Published:** 2024-03-27

**Authors:** Wenyuan Shang, Xiujun Hu, Xiaoman Lin, Shangru Li, Shuchang Xiong, Bingding Huang, Xin Wang

**Affiliations:** College of Big Data and Internet, Shenzhen Technology University, Shenzhen 518118, China; 202002020204@stumail.sztu.edu.cn (W.S.); 202002010413@stumail.sztu.edu.cn (X.H.); 202005010310@stumail.sztu.edu.cn (X.L.); 202002010319@stumail.sztu.edu.cn (S.L.); 202002010326@stumail.sztu.edu.cn (S.X.)

**Keywords:** nanobody, in silico screening, protein docking, COVID-19

## Abstract

Nanobodies (Nbs or VHHs) are single-domain antibodies (sdAbs) derived from camelid heavy-chain antibodies. Nbs have special and unique characteristics, such as small size, good tissue penetration, and cost-effective production, making Nbs a good candidate for the diagnosis and treatment of viruses and other pathologies. Identifying effective Nbs against COVID-19 would help us control this dangerous virus or other unknown variants in the future. Herein, we introduce an in silico screening strategy for optimizing stable conformation of anti-SARS-CoV-2 Nbs. Firstly, various complexes containing nanobodies were downloaded from the RCSB database, which were identified from immunized llamas. The primary docking between Nbs and the SARS-CoV-2 spike protein receptor-binding domain was performed through the ClusPro program, with the manual screening leaving the reasonable conformation to the next step. Then, the binding distances of atoms between the antigen–antibody interfaces were measured through the NeighborSearch algorithm. Finally, filtered nanobodies were acquired according to HADDOCK scores through HADDOCK docking the COVID-19 spike protein with nanobodies under restrictions of calculated molecular distance between active residues and antigenic epitopes less than 4.5 Å. In this way, those nanobodies with more reasonable conformation and stronger neutralizing efficacy were acquired. To validate the efficacy ranking of the nanobodies we obtained, we calculated the binding affinities (∆G) and dissociation constants (Kd) of all screened nanobodies using the PRODIGY web tool and predicted the stability changes induced by all possible point mutations in nanobodies using the MAESTROWeb server. Furthermore, we examined the performance of the relationship between nanobodies’ ranking and their number of mutation-sensitive sites (Spearman correlation > 0.68); the results revealed a robust correlation, indicating that the superior nanobodies identified through our screening process exhibited fewer mutation hotspots and higher stability. This correlation analysis demonstrates the validity of our screening criteria, underscoring the suitability of these nanobodies for future development and practical implementation. In conclusion, this three-step screening strategy iteratively in silico greatly improved the accuracy of screening desired nanobodies compared to using only ClusPro docking or default HADDOCK docking settings. It provides new ideas for the screening of novel antibodies and computer-aided screening methods.

## 1. Introduction

When it broke out in late 2019, COVID-19 was a major health problem worldwide, owing to the lack of effective treatment methods. With the rapid global spread of Omicron and other notable variants, the development of more potent antibodies and antiviral drugs has been a global concern. For instance, one approach is the yeast display VHH library construction method, which involves extracting peripheral blood lymphocytes from immunized alpacas [1]. Another method utilizes antibody modeling applications to construct nanobody models, focusing on regions such as the Fv region of an antibody [2].

There are three main proteins on the surface of the SARS-CoV-2 virus, namely spike protein (S), envelope protein (E), and membrane protein (M) [3,4]. Among them, S protein has a key role in virus entry into host cells and is an important target for SARS-CoV-2 infection. The S protein is a class of trimeric transmembrane glycoproteins, which has a full size of 1273 amino acid residues and consists of two functional subunits (S1 and S2) [5]. The S1 subunit is responsible for binding to the host cell receptor, and the S2 subunit can fuse with the viral and cellular membranes. The S1 subunit can be further divided into two functional domains, an N-terminal domain and a C-terminal domain (CTD) involving the receptor-binding domain (RBD), which is the main receptor used by SARS-Cov-2 for cell entry and directly binds to the receptor angiotensin-converting enzyme 2 (ACE2) on host cells [6,7]. The C-terminal domain is involved in the assembly and release of viral particles. Thus, S protein is a key molecule for viral infection of host cells by decorating the virion surface as a major antigen and inducing neutralizing antibody responses.

Tremendous progress in the structure and function of SARS-CoV-2 spike protein has been made since the initial outbreak of COVID-19. Some treatments have shown some benefits in certain patients to date [8,9,10]; however, there are no generally proven effective therapies for antivirals against SARS-CoV-2. Studies reported that specific monoclonal antibody therapy is an effective immunotherapy for SARS-CoV-2 infection [11], and vaccination is the most effective method for a long-term strategy for the prevention and control of COVID-19 [12,13]. However, the wide application of therapeutic monoclonal antibodies may be restricted by the high cost and limited capacity of manufacturing, as well as the problem of bioavailability. Screening, validation, and large-scale production of neutralizing antibodies take a long time to test, making it difficult to function at critical windows early in the outbreak. Herein, the engineering of nanobodies emerges as a positive solution.

Nanobodies (Nbs) are special types of antibodies with no light chains (L) in their structure and are composed of only heavy chains (H). They possess a significantly smaller molecular weight of only 14 kDa, which is approximately one-tenth the size of a conventional antibody (Figure 1). They are derived from the peripheral blood of Camelidae family animals, including alpacas. The peculiar properties of Nbs include nanoscale size, stability, high affinity and specificity, water solubility, and high tissue penetration [14]. These Nbs have attracted considerable interest, and the applications of Nbs depend on reliable, cost-effective, and high-volume production. These unique properties of Nbs over conventional antibodies make them powerful therapies against SARS-CoV-2 [15,16]. Effective nanobodies have been successfully generated in biological experiments [1,17], which could be used for targeting the SARS-CoV-2 virus for the diagnosis and treatment of COVID-19. Computational studies on Nbs targeting SARS-CoV-2 S protein have attracted enormous attention [18]. However, there is a lack of in silico screening techniques to promptly identify a large number of nanobodies that can effectively neutralize the SARS-CoV-2 virus.

In this paper, we highlight an iterative screening strategy: the flowchart in Figure 2 depicts the methodology used to predict the molecular docking and binding free energy between nanobodies and the SARS-CoV-2 spike protein RBD using ClusPro [19] and HADDOCK [20] molecular docking servers with the algorithm of NeighborSearch, which predicted the active binding sites for HADDOCK to calculate the final results. Notably, we identified the active residues on both the antigen and nanobody and subsequently integrated them into HADDOCK to enhance the performance of the in silico screening.

## 2. Results

We preprocessed the Nbs and spike proteins using PyMOL [21] and removed unwanted molecules such as water and chloride ions, as well as impurities such as small-molecule ligands or other proteins. We also checked structural issues and then used ClusPro to dock them. Multiple sets of predictive conformations were obtained by docking each pair of spike proteins and nanobodies. However, some predicted conformations did not exhibit nanobody binding within the RBD region of the spike protein, which is a critical area to screen for nanobodies capable of effectively competing with ACE2 for binding to the RBD domain [22] (Figure 3).

Further screening was then performed using HADDOCK, active residues were added for precise localization, and the conformation of the binding site within the RBD region was obtained. The active residues are situated on the interface of interaction between the nanobody and S protein. We set the distance threshold (4.5 Å) between the molecules at the antigen–antibody interface to clarify if these regions are active residues. Within this 4.5 Å distance range, the residue interactions between protein chains are regarded as strong interactions. Active residues play a crucial role in maintaining the stability of the protein complex and conferring binding specificity [23]. Figure 4 shows the predicted active residues in the docking of the S protein and nanobody, indicating the positions and corresponding residue IDs of active residues.

Finally, HADDOCK docking was completed, and each predicted conformation was assigned an evaluation score. The HADDOCK score is calculated using a linear function that combines various energies and buried surface area [20]. The lower the score, the better the conformation. Moreover, with the ClusPro pre-screening step, the binding sites of the antibodies are reasonably located within the RBD region. Based on the best score ranking from each predicted conformation, we selected the top 30 excellent nanobodies. In Table 1, we listed the source PDBID, the best HADDOCK score, and the HADDOCK score error range for each nanobody.

To validate our findings, we used Spearman correlation to assess the correlation between the rankings of nanobodies and the number of mutation-sensitive positions they exhibited (Figure 5). In PRODIGY [24], our findings indicate that the top-ranking nanobodies also displayed concurrent high binding affinities (Table 2).

## 3. Discussion

COVID-19 is an infectious disease caused by the novel coronavirus known as SARS-CoV-2. As the COVID-19 pandemic erupted globally in late 2019, it caused tremendous impacts on human health, economies, and other various aspects in many countries. As of May 2023, there have been over 760 million confirmed cases and more than 69,000 deaths [25].

The rapid spread of the COVID-19 pandemic can be attributed to the primary modes of transmission of the SARS-CoV-2 virus, which are airborne and contact transmission. The virus can spread through respiratory secretions, droplets, and aerosols. Simply being in contact with an individual infected with SARS-CoV-2 poses a risk of transmission. People may experience various symptoms, including fever, cough, shortness of breath, fatigue, muscle pain, sore throat, loss of taste or smell, and others. Severe cases may require hospitalization and even respiratory support with ventilators.

The COVID-19 pandemic has had profound implications. Over the past four years, extensive research has been conducted on various antiviral antibodies or drugs for the prevention and treatment of COVID-19. To date, the World Health Organization’s International Clinical Trials Registry Platform (https://clinicaltrials.gov/ct2 (accessed on 20 March 2024)) has listed 9581 ongoing and completed COVID-19 studies. However, only half of these study statuses indicate ‘completed’, highlighting the ongoing research needs regarding the availability and accessibility of antibodies. During the early stages of the COVID-19 pandemic, researchers extensively collected and screened neutralizing antibodies (NAbs), obtained from convalescent patients [26]. However, the rapid and iterative emergence of variants, such as Omicron and Delta, has adversely affected the therapeutic efficacy of natural antibodies.

In the context of the continuously emerging highly contagious variants, in silico approaches have played a crucial role by offering a rapid and cost-effective alternative to the widely used trial-and-error approach in experimental research. This allows researchers to efficiently study mutation sites to discover design strategies to overcome immune evasion and screen for effective antibodies while saving valuable time. In recent times, in silico screening methods have not only been employed in docking studies for chemical drug design [27] but have also been successfully utilized to identify various effective immunotherapeutic agents. These agents include vaccines [28], chimeric antibodies, and nanobodies [29], which have been utilized for the treatment of various infectious and inflammatory diseases in humans. Moreover, nanobodies have attracted significant attention because of their small size, ability to recognize hidden epitopes, and ease of production. Several of these nanobodies have been developed for identifying or inhibiting viruses, such as hepatitis E virus (HEV) [30], influenza virus, and respiratory syncytial virus. The use of nanobodies may be one of the most effective approaches for protecting against COVID-19.

For the reasons above, we hope to combine the advantages of in silico screening methods and nanobodies and extend the design of this computational screening method to identify a variety of effective antibodies, especially nanobodies. This study aimed to improve the accuracy of docking predictions for the spike protein and nanobody complex by introducing HADDOCK for iterative filtering of ClusPro pre-screened docking results. The calculated active residues were used as a constraint output to refine the ClusPro docking predictions and reduce computational complexity.

This approach not only increased the accuracy of ClusPro/HADDOCK docking but also addressed the issue of how to filter out unrealistic binding sites that might result from pure rigid docking. Then, the HADDOCK score was used as a comprehensive evaluation index to compare the quality and accuracy of different docking conformations, with a lower score indicating a more stable and reliable docking conformation.

In future studies, we aim to employ this iterative filtering approach to dock a wider range of viral variants and identify high-quality nanobodies to specific pathogens.

In the next step, we would also integrate this iterative filtering module into our NanoLAS online platform [31], providing an integrated and efficient service for screening nanobodies. In NanoLAS 2.0, we will incorporate additional rigid docking programs, including ZDOCK [32], GRAMM-X [33], pyDOCK [34], and FTDock [35]. These programs exhibit comparable performance to ClusPro and can serve as alternative options for the initial screening step. After the initial rigid docking, if further precise screening of higher-quality nanobodies is required, the user can directly input the output of the first screening step into the next module. The module will calculate the active residues of the antigen–antibody binding site and use it for more precise and targeted screening via algorithms such as HADDOCK or RosettaDOCK [36].

Furthermore, the residue IDs of binding sites on the spike protein chain in the models may provide important information on potentially binding epitopes. In our experience, it is crucial to find nanobody structures that match the epitopes with the active residue information. We expect our method to be extended to nanobodies screening of other diseases, especially in related neoplastic diseases such as lung cancers and brain tumors.

Currently, when screening for reasonable conformations from ClusPro, we still rely on manual selection based on our experience. Manual screening could also be replaced by programs with appropriate algorithms in the future. This technological progress will further standardize the screening flow and improve efficiency.

## 4. Materials and Methods

### 4.1. Data Mining

In this study, we searched for PDB files containing nanobody complexes related to SARS viruses in the RCSB [37] database (https://www.rcsb.org/ (accessed on 20 March 2024)). We selected nanobodies obtained through biological experiments, especially utilizing cryo-electron microscopy, as these experimental-derived nanobodies are more reliable in terms of stability and affinity when compared to nanobodies synthesized using modeling software. Additionally, we also retrieved a PDB file containing the natural spike protein of the SARS-CoV-2 from the RCSB database (PDB id 7ddn [38]).

### 4.2. Molecular Docking and Determination of Biophysical Interactions

#### 4.2.1. Pre-Docking with ClusPro 2.0

Molecular docking [39] was used to determine protein–protein interactions, which is a useful and validated technique for antibody screening.

We first employed ClusPro 2.0 [19,40,41,42] (https://cluspro.org/ (accessed on 20 March 2024), Vajda Lab and ABC Group, Boston University (Boston, MA, USA) and Stony Brook University (Stony Brook, NY, USA)) for docking. ClusPro is a computational tool for molecular docking of diverse molecular interactions, including protein–protein, protein–nucleic acid, and protein–small molecule complexes. Information including interaction force, binding free energies, and combined modes between molecules can be predicted by molecular mechanics and Monte Carlo simulation techniques. For each antigen–antibody complex, ClusPro retained 1000 lowest energy models and evaluated the top 30 models (centers of the 30 largest clusters) using the DockQ program [43]. When some complexes have fewer than 30 clusters, a DockQ result of 0.23 (acceptable) and above (better) would be considered a good solution (ClusPro considers a DockQ result of 0.23 and above as a good solution [40]). The top 15 predictions of each docking conformation were considered for the next manual selection step.

#### 4.2.2. Manual Screening Step

However, when only ClusPro was employed, we noticed that some predicted antibody binding sites were not within the virus’s RBD. Although they displayed good stability, these protein pairing confirmations are not useful and should be filtered out manually. So, we added a manual screening step hereafter ClusPro docking to filter out the docking results that docked outside the virus’s RBD region.

### 4.3. Identifying Active Residues

The stability of antigen–antibody binding can be evaluated by factors such as the interface area of interaction, hydrogen bonds, van der Waals interactions, etc., at the interface. In addition, the distance between atoms at the antigen–antibody interface is also an important indicator for evaluating binding stability. Bio.PDB.NeighborSearch is a powerful Python module that is part of the Biopython [44] library. It seamlessly integrates with other protein structure analysis tools and data structures. It can interact with objects such as Structure, Model, Chain, and Residue from Bio.PDB, providing a fast and efficient method to search for neighboring atoms or residues in protein structures.

The Bio.PDB.NeighborSearch accomplishes rapid nearest neighbor search by constructing k-d trees [45]. A k-d tree (k-dimensional tree) is a binary tree data structure used for partitioning and organizing points in k-dimensional space. In the context of protein structures, each atom can be represented as a k-dimensional vector, where k is the dimensionality of the space (typically 3, representing three-dimensional spatial coordinates). The k-d tree divides the space into multiple hyper-rectangular regions, where each node represents a hyper-rectangular region and is stored in the tree nodes.

The algorithm steps are as follows: Firstly, a partitioning dimension is selected (typically the dimension exhibiting the highest variance), and a partitioning value is chosen to segregate the data into left and right subtrees. This partitioning process is recursively applied to the left and right subtrees until each leaf node exclusively encompasses a singular atom or residue.

The search commences from the root node. By comparing the query point with the partitioning value of the current node, the direction of the search, whether within the left subtree or the right subtree, is determined. The same search process is repeatedly executed within the selected subtree until the nearest neighbor atom is identified, and finally, the program returns the neighboring atoms found.

For antigen–antibody, we used the NeighborSearch algorithm to traverse the atoms of both chains. For each pair of atoms, the program uses the NeighborSearch algorithm to find the distance between chains. If the distance found is not greater than the set threshold, it predicates that the atom in the antibody chain is a “close atom”. The residue, composed of multiple atoms, is the fundamental structural unit of a protein. We identified key residues as active residues at the binding interfaces of the antigen–nanobody from the close atoms and saved its ID for the HADDOCK step.

(The code of active residue identification can be downloaded at: https://github.com/WangLabforComputationalBiology/SARS-COV-nanobody-screening (accessed on 20 March 2024)).

### 4.4. Further Docking with HADDOCK 2.4

We employed HADDOCK [20,46] for the next step iterative screening, which can perform rigid, semi-flexible, and fully flexible docking (https://wenmr.science.uu.nl/haddock2.4/ (accessed on 20 March 2024), Bonvinlab, Utrecht University, Utrecht, The Netherlands). The screening steps of HADDOCK provide several advantages for further docking and screening.

Firstly, HADDOCK incorporates information from identified or predicted protein interfaces, utilizing ambiguous interaction restraints (AIRs) to guide the docking process. This allows for a more informed and targeted exploration of the protein–protein interaction space. Additionally, HADDOCK enables the inclusion of specific unambiguous distance restraints, such as those derived from Mass Spectrometry cross-links, as well as various other experimental data sources, including NMR residual dipolar couplings and cryo-EM maps. This integration of diverse experimental data enhances the accuracy and reliability of the docking predictions [47].

Moreover, by incorporating active residues to constrain the prediction range, the accuracy of docking prediction results can be improved, and the computational complexity can be reduced.

By first using ClusPro for primary screening, we were able to quickly select reasonable conformations to calculate their active residue. Then, with the active residue information acquired by Method C, HADDOCK could be used to further evaluate the performance of the docking more precisely.

### 4.5. Determination of Binding Affinity and Stability Validation of Screening

The energy of protein–protein interactions between the spike protein fragment and nanobodies is an important factor in evaluating the stability of different nanobodies upon binding. Binding affinity represents the strength of protein–protein interactions and is associated with the cellular functions of these proteins. This binding energy can be quantified as a physicochemical parameter known as the dissociation constant (Kd), which provides a quantitative measure of the affinity between a ligand and its target molecule. Kd represents the equilibrium concentration of the unbound ligand when the ligand–target complex is in dynamic equilibrium. A smaller Kd value indicates a higher binding affinity, suggesting a more stable and favored binding interaction.

In parallel, the thermodynamic parameter Gibbs free energy (ΔG) is directly related to Kd through the equation ΔG = −RT ln(Kd), where R is the gas constant, and T is the temperature [48]. Therefore, a lower Kd value corresponds to a more negative ΔG, indicating a stronger binding affinity and a more thermodynamically stable formation of the ligand–target complex.

The prediction of binding affinity between nanobodies and the spike protein was performed using the PRODIGY server, which is based on an intermolecular contacts prediction method for protein–protein complex binding affinity.

In addition to evaluating the binding affinity in diverse conformations, assessing the robustness of nanobodies is a crucial factor in determining their suitability for industrial production. While protein mutations can have beneficial effects on physicochemical or biological properties, they can also result in structural variations and loss of function. Therefore, estimating the stability changes caused by all possible point mutations can provide an additional validation of the exceptional quality of the screened nanobodies. For prediction, we used the ‘Calculate a mutation sensitivity profile’ module on the MAESTROweb [49] server, which shares similarities with the PoPMuSiC [50] web server.

In the MAESTROweb server, the stability changes resulting from point mutations are denoted as ΔΔ*G* (*s_w_*, *s_m_*), which are used to estimate the folding free energy change when residue *s_w_* is mutated to *s_m_*. It is calculated based on the wild-type protein structure and a set of energy functions:$$∆∆G_{p}=\sum_{i=1}^{13}\alpha_{i}\left(A\right)∆∆W_{i}+\alpha_{14}\left(A\right)∆V_{+}+\alpha_{15}\left(A\right)∆V_{-}+\alpha_{16}\;\left(A\right)\;$$

The coefficients *α_i_* depend on the solvent accessibility *A* of the wild-type amino acid *s_w_*. ΔΔ*W_i_* is a linear combination of 13 statistical potentials, which describe the torsional angles of amino acid type *s*, defined backbone conformation *t*, solvent accessibility *a*, and the correlation between the spatial distance *d* of the average geometric centers of each residue pair and their side chains. Δ*V*_±_ is related to the difference in volume between the mutant amino acid and the wild-type amino acid: Δ*V* = *Vm* − *Vw*. They are defined as Δ*V_±_* = Δ*VH* (± Δ*V*), where H represents the Heaviside function. They offer a depiction of the effects associated with the formation of a cavity (when Δ*V* < 0) or the accommodation of a larger side chain within the protein structure (when Δ*V* > 0).

For each position *i* in the protein, ΔΔ*G* can be calculated for all 19 possible mutations. (ΔΔ*G_pred_* < 0.0 indicates a confirmed stabilizing mutation.) Mutations occurring at positions with lower ΔΔ*G_pred_* values have a lesser impact, indicating a reduced sensitivity to mutation. We conducted a statistical analysis of the MAESTROweb results for each nanobody, counting the number of sensitive positions where ΔΔ*G_pred_* exceeded 1.8 (kcal/mol). Additionally, the MAESTROweb server provides a visualization service, which can show us the impact of each mutation at all potential positions (Figure 6).

## 5. Conclusions

Compared to using only ClusPro docking or only default HADDOCK docking settings, this three-step approach provided a more comprehensive and efficient way of analyzing the molecular interactions between the S protein and the nanobodies. This strategy not only provides a valuable tool for the selection of nanobodies but also presents new ideas for the screening of novel antibodies and the development of computer-aided screening methods. The iterative process and the incorporation of multiple algorithms with customized settings can be applied to other antibody screening projects and may lead to the identification of more effective therapeutic antibodies.

Moreover, we conducted a comprehensive investigation on the stability of nanobodies in relation to their rankings. To accomplish this, we used the MAESTROWeb server to predict the stability changes resulting from point mutations in nanobodies. We performed a statistical analysis to quantify the number of potential mutation sites. Our analysis demonstrated a significant correlation between the number of mutation-sensitive sites that exhibited significant energy fluctuations and the rankings of the nanobodies. In conclusion, our three-step screening strategy effectively identifies nanobodies with superior mutation resistance, rendering them highly suitable for practical production applications.

## Figures and Tables

**Figure 1 pharmaceuticals-17-00424-f001:**
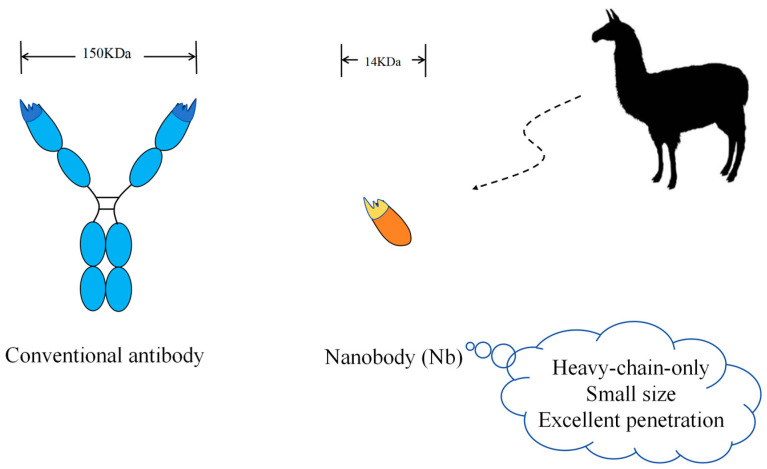
Nanobodies harbor advantageous features compared to conventional antibodies since they are composed of only a single heavy chain instead of the typical two heavy and two light chains found in conventional antibodies.

**Figure 2 pharmaceuticals-17-00424-f002:**
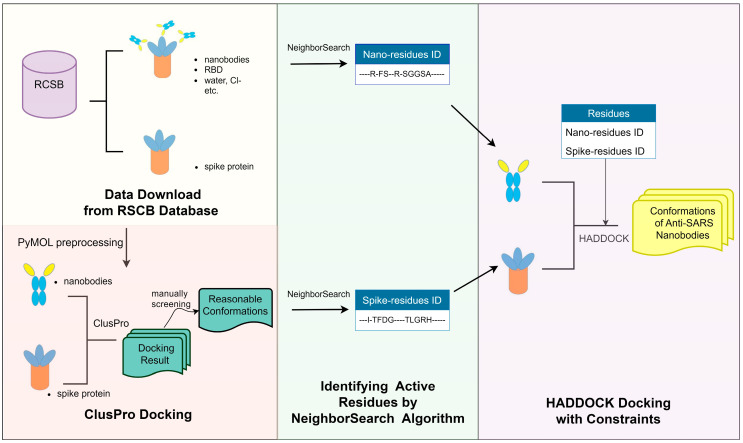
The flowchart of the three-step in silico screening strategy. (**Left**) panel: the first screening step by ClusPro. (**Middle**) panel: identifying active residues of spike by the NeighborSearch algorithm. (**Right**) panel: further screening by HADDOCK with active residue information.

**Figure 3 pharmaceuticals-17-00424-f003:**
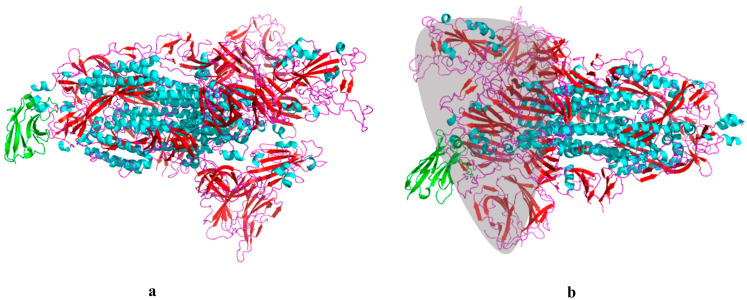
A comparison of two ClusPro-predicted models for nanobody binding sites, nanobodies were colored in green and the spike protein of SARS- CoV-2 was in three other colors, representing different structures. (**a**) Nanobodies bind outside the SARS- CoV-2 RBD domain. (**b**) Nanobodies bind within the SARS-CoV-2 RBD domain (shaded area).

**Figure 4 pharmaceuticals-17-00424-f004:**
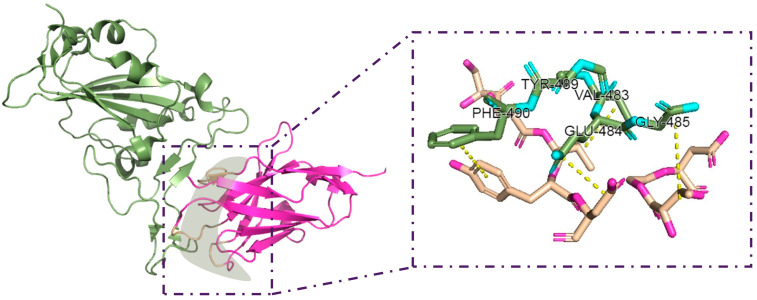
Visualization of program-computed active residues by the NeighborSearch algorithm. To distinguish chains more clearly, we set them to different colors. The dotted line represents for the distance between residues.

**Figure 5 pharmaceuticals-17-00424-f005:**
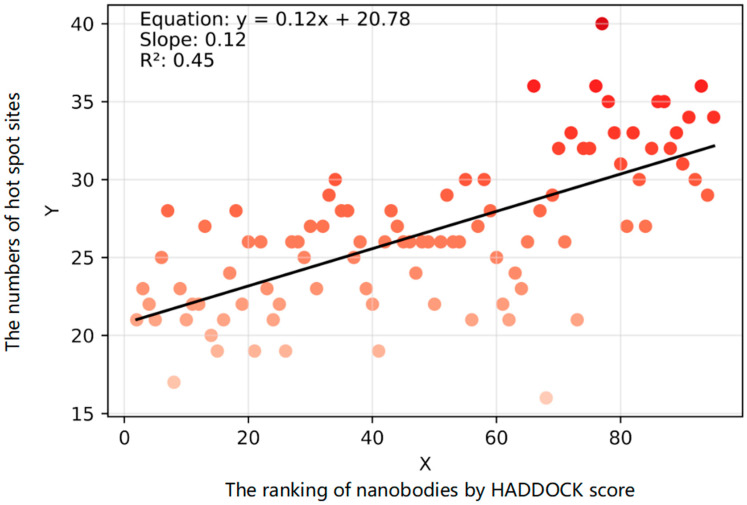
Correlation plot depicting the predicted count of mutation-sensitive sites (hot spot sites) and the ranking order of the screened nanobodies. The y-axis is the number of hot spot sites, and the x-axis is the ranking of nanobodies by HADDOCK score.

**Figure 6 pharmaceuticals-17-00424-f006:**
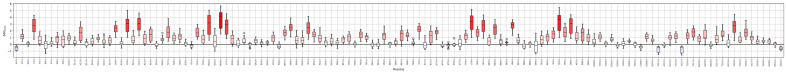
Mutation sensitivity profile of Nbs (PDBID: 7tpr). The vertical axis represents ΔΔG, and the horizontal axis depicts amino acid residues, with red color represents residues with more mutations.

**Table 1 pharmaceuticals-17-00424-t001:** Results of nanobodies screened by HADDOCK scores. (Full list can be downloaded at https://www.nanolas.cloud/download/Screening%20Nano%20Full.csv (accessed on 20 March 2024)).

Source	Nanobody Name	Best HADDOCK Score	Error Range
(PDB ID)			
7tpr	Nanobody 8A2	−175.8	±5.9
7tpr	Nanobody 7A3	−162.2	±5.1
7rby	Nanobody nb-112	−154	±2.9
7rxd	Nb_RBD	−153.9	±3.3
7vq0	Nanobody P86	−149.4	±13.2
7vnb	n3113	−148.8	±1.9
8cyb	Nanobody 1–8	−148.7	±1.2
7a25	Sybody 23	−148	±5.2
8cy7	Nanobody 2–38	−143.9	±4.8
7r4r	Nanobody 1.10	−141.5	±0.6
7r4q	Nanobody 1.29	−141.3	±3.0
7x4i	Nanobody aSA3	−140.1	±2.6
8cyc	Nanobody 2–34	−137.7	±9.2
7b14	Nanobody	−137.7	±4.3
8bev	Nanobody W25	−137.5	±4.6
7whi	Bn03_nano2	−136.1	±3.0
8cy9	Nanobody 1–23	−133.7	±4.3
7x2j	Nb70	−133.5	±2.2
7whi	Bn03_nano1	−133.1	±5.9
7zf4	Nanobody F2	−132.5	±3.4
7wpf	Nanobody	−132.2	±4.6
8dqu	Nanobody	−130.9	±2.2
8cya	Nanobody 2–67	−130.8	±3.2
7xrp	C5G2 nanobody	−129.4	±7.7
8cwu	VHH 1–21	−128.3	±4.8
7xod	Nanobody	−128.1	±7.9
7voa	Nanobody	−127.9	±2.3
7wd2	Nanobody	−125	±3.7
7r98	Nanobody B6	−124.1	±4.7
7z85	Nanobody H11-B5	−123.6	±3.2

**Table 2 pharmaceuticals-17-00424-t002:** Affinity of top-ranking nanobodies against spike protein.

Source PDB	ΔG (kcal mol^−1^)
8cy7	−15.8
7tpr2	−15.4
8cyb	−13.6
7tpr	−13.4
7vnb	−13.1

## Data Availability

Data is contained within the article.

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
