# Peer review of "Iterative In Silico Screening for Optimizing Stable Conformation of Anti-SARS-CoV-2 Nanobodies"

_pharmaceuticals, 2024, doi:10.3390/ph17040424_

Round 1
Reviewer 1 Report
Comments and Suggestions for Authors
As part of the presented article, the authors conducted research in the field of developing new approaches to the search for effective antibodies for the treatment of viral diseases, in particular, this particular study examines the search for antibodies produced by the camelid family to the S-protein of covid-19. ,The prospects for the generation of antibodies by the camelid family, as well as the effectiveness of methods developed in silico, are shown. Despite the fact that I believe that the number of examples on which the model was trained and tested is insufficient. More examples of antibodies that act as positive and negative controls are needed in order to draw global conclusions. And also, I believe that the authors should more fully describe the capabilities of the method in relation to other types of antibodies, as well as other types of protein, both covid19 and other viral infections. Nevertheless, I believe that this article can still be recommended for publication, since it is of significant interest to both specialists in the field of machine screening and practicing molecular biologists. But subject to a more clear statement of the problem and a description of further prospects and areas of application of the proposed methods.
Author Response
Reviewer 1:
As part of the presented article, the authors conducted research in the field of developing new approaches to the search for effective antibodies for the treatment of viral diseases, in particular, this particular study examines the search for antibodies produced by the camelid family to the S-protein of covid-19. The prospects for the generation of antibodies by the camelid family, as well as the effectiveness of methods developed in silico, are shown. Despite the fact that I believe that the number of examples on which the model was trained and tested is insufficient. More examples of antibodies that act as positive and negative controls are needed in order to draw global conclusions. And also, I believe that the authors should more fully describe the capabilities of the method in relation to other types of antibodies, as well as other types of protein, both covid19 and other viral infections. Nevertheless, I believe that this article can still be recommended for publication, since it is of significant interest to both specialists in the field of machine screening and practicing molecular biologists. But subject to a more clear statement of the problem and a description of further prospects and areas of application of the proposed methods.
Revision:
We sincerely appreciate the valuable comments. We added discussion of the other drug research methods and advantages of in silico screening methods. In the last four paragraphs of the Discussion section, we stated further prospects and application areas of our screening method.
We also added more data in the address https://www.nanolas.cloud/download/Screening%20Nano%20Full.csv (below Table 1), now there are 118 entries in total with 23 newly added below Top 50.

Reviewer 2 Report
Comments and Suggestions for Authors
Comments
Type of paper: Research
ID: Parhamceuticals-2909456
Journal: Pharmaceuticals
The manuscript aims to demonstrate the value of an in silico approach in the screening of potential nanobodies tackling COVID-19. Overall, it is a well-written paper with sufficient detail. Nevertheless, some aspects can be improved and are presented below.
Abstract
Very well written and informative. Be careful with the abbreviations (nanobodies) and please specify the COVID strain used in the study.
Line 10: I am not sure if it is accurate to say that “variable region of these nanobodies has special and unique characteristics, (…). Better replace variable region for Nbs or VHHs.
Line 12: Nbs instead of nanobodies
Line 12: viruses and other pathologies…
Line 13: “defeat” is too strong…
Line 15: Nbs
Keywords
And the word COVID-19?
Introduction
Well written and clear.
Line 46: antibodies and antiviral drugs.
Line 48: It would be nice to have a scheme of this since the description is so well done
Line 67: It would be interesting to add a sentence discussing a little the problem of rapid production of antibodies in case of a breakthrough. Herein, the engineering of nanobodies is also a positive solution.
Line 70: a figure comparing the advantages and disadvantages of nanobodies over antibodies would be a good add-on.
Line 73: add molecular weight as a comparison to the “normal” antibodies size
Line 74: “suitable source et al”???
Line 83: The strain of the COVID-19 virus used is important in this context or not? If it is, please add the information concerning that.
Methods
Methods are well written, with sufficient details and informative. In which step did you perform simulations? Did you not consider it?
Line 100: why are they more reliable? Comment on that, please.
Line 101: all variants?
Line 109: “Massive information”?
Results
Well presented.
Line 220: nanobodies… abbreviations…
Line 252: Table I? Or table 1?
Line 255: in some cases, authors write Figure and in others Fig. Please, check it.
Line 257: Table II? Or table 2?
Line 244: “The vertical axis represents ΔΔG, 244 and the horizontal axis depicts amino acid residues.” Is this correct and meant to be here or in Figure 5?
Line 260: Where is Figure 4?
Table 267: And the information on the “dissociation constant (Kd)”?

Author Response
Please see the attachement below.

Round 2
Reviewer 2 Report
Comments and Suggestions for Authors
The authors replied successfully to all my comments